# LOCAL MINIMA IN TRAINING OF DEEP NETWORKS

**Grzegorz Świrszcz, Wojciech Marian Czarnecki & Razvan Pascanu**
DeepMind
London, UK
`{swirszcz,lejlot,razp}@google.com`

## ABSTRACT

There has been a lot of recent interest in trying to characterize the error surface of deep models. This stems from a long standing question. Given that deep networks are highly nonlinear systems optimized by local gradient methods, why do they not seem to be affected by *bad* local minima? It is widely believed that training of deep models using gradient methods works so well because the error surface either has no local minima, or if they exist they need to be close in value to the global minimum. It is known that such results hold under very strong assumptions which are not satisfied by real models. In this paper we present examples showing that for such theorem to be true additional assumptions on the data, initialization schemes and/or the model classes have to be made. We look at the particular case of finite size datasets. We demonstrate that in this scenario one can construct counter-examples (datasets or initialization schemes) when the network does become susceptible to bad local minima over the weight space.

## 1 INTRODUCTION

Deep Learning (LeCun et al., 2015; Schmidhuber, 2015) is a fast growing subfield of machine learning, with many impressive results. One particular criticism often brought up against this family of models is the fact that it relies on non-convex functions which are optimized using local gradient descent methods. This means one has no guarantee that the optimization algorithm will converge to a meaningful minimum or even that it will converge at all. However, this theoretical concern seems to have little bearing in practice.

In Dauphin et al. (2013) a conjecture had been put forward for this based on insights from statistical physics which point to the scale of neural networks as a possible answer. The claim is that the error structure of neural networks might follow the same structure as that of random Gaussian fields which have been recently understood and studied in Fyodorov & Williams (2007); Bray & Dean (2007). The critical points of these functions, as the dimensionality of the problem increases, seem to have a particularly friendly behaviour where local minima align nicely close to the global minimum of the function. Choromanska et al. (2015) provides a study of the conjecture by mapping deep neural models onto spin-glass ones for whom the above structure holds. These work has been extended further (see Section 2 for a review of the topic).

We believe many of these results do not trivially extend to the case of finite size datasets/finite size models. The learning dynamics of the neural network in this particular case can be arbitrarily bad. Our assertions are based on constructions of counterexamples that exploit particular architectures, the full domain of the parameters and particular datasets.

## 2 LITERATURE REVIEW

One view, that can be dated back to Baldi & Hornik (1989), about why the error surface of neural networks seems well behaved is the one stated in Dauphin et al. (2013). We would refer to this hypothesis as the "no bad local minima" hypothesis. In Baldi & Hornik (1989) it is shown that an MLP with a single *linear* intermediate layer has *no* local minima, only saddle points and a global minimum. This intuition is carried further by Saxe et al. (2014; 2013), where deep linear models are studied. While, from a representational perspective, deep linear models are not useful, the hope is

that the learning dynamics of such models can be mathematically understood while still being rich enough to mirror the dynamics of nonlinear networks. The findings of these works are aligned with Baldi & Hornik (1989) and suggest that one has only to go through several saddles to reach a global minimum.

These intuitions are expressed clearly for generic deep networks in Dauphin et al. (2013). The key observation of this work is that intuitions from low-dimensional spaces are usually misleading when moving to high-dimensional spaces. The work makes a connection with deep results obtained in statistical physics. In particular Fyodorov & Williams (2007); Bray & Dean (2007) showed, using the Replica Theory (Parisi, 2007), that random Gaussian error functions have a particular friendly structure. Namely, if one looks at all the critical points of the function and plots error versus the (Morse) index of the critical point (the number of negative eigenvalues of the Hessian), these points align nicely on a monotonically increasing curve. That is, all points with a low index (note that every minimum has this index equal to 0) have roughly the same performance, while critical points of high error implicitly have a large number of negative eigenvalue which means they are saddle points.

These observations align also with the theory of random matrices (Wigner, 1958) which predicts the same behaviour for the eigenvalues of a random matrix as the size of the matrix grows. The claim of Dauphin et al. (2013) is that same structure holds for neural network as well, when they become large enough. Similar claim is put forward in Sagun et al. (2014). The conjecture is very appealing as it provides a strong argument why deep networks end up performing not only well, but also reliably so. Choromanska et al. (2015) provides a study of the conjecture that rests on recasting a neural network as a spin-glass model.To obtain this result several assumptions need to be made, which the authors of the work, at that time, acknowledged that were not realistic in practice. The same line of attack is taken by Kawaguchi (2016).

Goodfellow et al. (2016) argues and provides empirical evidence that while moving from the original initialization of the model along a straight line to the solution (found via gradient descent) the loss seems to be only monotonically decreasing, which speaks towards the apparent convexity of the problem. Soudry & Carmon (2016); Safran & Shamir (2015) also look at the error surface of the neural network, providing theoretical arguments for the error surface becoming well-behaved in the case of over-parametrized models.

A different view, presented in Lin & Tegmark (2016); Shamir (2016), aligned with this work, is that the underlying easiness of optimizing deep networks does not simply rest just in the emerging structures due to high-dimensional spaces, but is rather tightly connected to the intrinsic characteristics of the data these models are run on.

## 3 THEORETICAL EXAMPLES AND RESULTS

We propose to analyze the error surface of rectified MLPs on finite datasets. The approach we take is a construction one. We build examples of datasets and model initializations that result in bad learning dynamics. In most examples we use ReLU units, as they are the most commonly used activation functions for both classification and regression tasks (e.g. in deep reinforcement learning (Mnih et al., 2015; 2016)). It is worth noting, however, that the phenomena we are demonstrating are not limited in nature to ReLU setup and they manifest themselves also for non-saturating activation functions like sigmoids.

### 3.1 LOCAL MINIMA IN A RECTIFIER-BASED REGRESSION

In Figure 1 we present 3 examples of local minima for regression using a single layer with 1, 2 and 3 hidden rectifier units on 1-dimensional data. For the sake of simplicity of our presentation we will describe in detail the case with 1 hidden neuron, the other two cases can be treated similarly. In case of one hidden neuron the regression problem becomes

$$\arg\min_{w,b,v,c} \mathcal{L}(w,b,v,c) = \sum_{i=1}^{n} \left( v \cdot \text{ReLU}(wx_i + b) + c - y_i \right)^2 . \tag{1}$$

Consider a dataset $\mathcal{D}_1$ (see Figure 1 (a)):

$$(x_1, y_1) = (5, 2), (x_2, y_2) = (4, 1), (x_3, y_3) = (3, 0), (x_4, y_4) = (1, -3), (x_5, y_5) = (-1, 3).$$

**Proposition 1.** *For the dataset $\mathcal{D}_1$ and $\mathcal{L}$ defined in Equation (1) the point $v = 1, b = -3, w = 1, c = 0$ is a local minimum of $\mathcal{L}$, which is not a global minimum.*

*Proof.* See Appendix B.4. □

**Remark 1.** *The point $(1, -3, 1, 0)$ is a minimum, but it is not a "strict" minimum - it is not isolated, but lies on a 1-dimensional manifold at which $\mathcal{L} \equiv 18$ instead.*

**Remark 2.** *One could ask whether* blind spots *are the only reasons for bad behaviour of rectifier nets. The answer is actually negative, and as following examples show – they can be completely absent in local optima, at the same time exisiting in a global solution!*

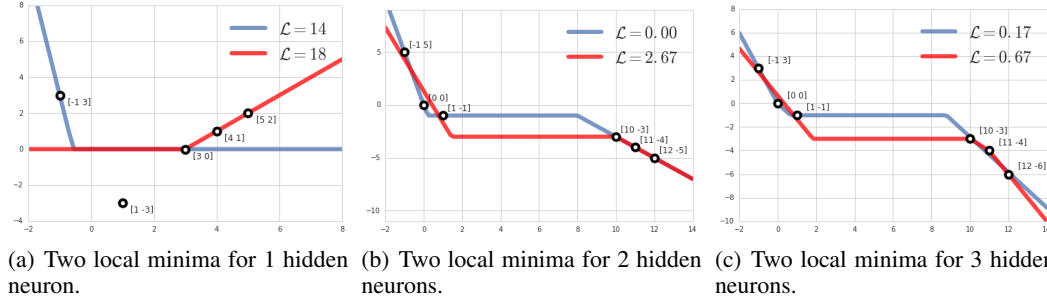

(a) Two local minima for 1 hidden neuron.  (b) Two local minima for 2 hidden neurons.  (c) Two local minima for 3 hidden neurons.

Figure 1: Local minima for ReLU-based regression. Both lines represent local optima, where the blue one is better than the red one.

**Proposition 2.** *Let us consider a dataset $\mathcal{D}_2$ with $d = 1$, given by points $(x_1, y_1) = (-1, 5), (x_2, y_2) = (0, 0), (x_3, y_3) = (1, -1), (x_4, y_4) = (10, -3), (x_5, y_5) = (11, -4), (x_6, y_6) = (12, -5)$ (Figure 1 (b)). Then, for a rectifier network with $m = 2$ hidden units and a squared error loss the set of weights $\mathbf{w} = (-5, -1), \mathbf{b} = (1, -8), \mathbf{v} = (1, -1), c = -1$ is a global minimum (with perfect fit) and the set of weights $\mathbf{w} = (-3, -1), \mathbf{b} = (4 + \frac{1}{3}, -10), \mathbf{v} = (1, -1), c = -3$ is a suboptimal local minimum.*

*Proof.* Analogous to the previous one. □

Maybe surprisingly, the global solution has a *blind spot* - all neurons deactivate in $x_3$. Nevertheless, the network still has a 0 training error. This shows that even though blind spots were used previously to construct very bad examples for neural nets, sometimes they are actually needed to fit the dataset.

**Proposition 3.** *Let us consider a dataset $\mathcal{D}_3$ with $d = 1$, given by points $(x_1, y_1) = (-1, 3), (x_2, y_2) = (0, 0), (x_3, y_3) = (1, -1), (x_4, y_4) = (10, -3), (x_5, y_5) = (11, -4), (x_6, y_6) = (12, -6)$ (Figure 1 (c)). Then, for a rectifier network with $m = 3$ hidden units and a squared error loss the set of weights $\mathbf{w} = (-1.5, -1.5, 1.5), \mathbf{b} = (1, 0, -13 - \frac{1}{6}), \mathbf{v} = (1, 1, -1), c = -1$ is a better local minimum than the local minimum obtained for $\mathbf{w} = (-2, 1, 1), \mathbf{b} = (3 + \frac{2}{3}, -10, -11), \mathbf{v} = (1, -1, -1), c = -3$.*

*Proof.* Completely analogous, using the fact that in each part of the space linear models are either optimal linear regression fits (if there is just one neuron active) or perfect (0 error) fit when two neurons are active and combined. □

Note that again that the above construction is not relying on the *blind spot* phenomenon. The idea behind this example is that if, due to initial conditions, the model partitions the input space in a suboptimal way, it might become impossible to find the optimal partitioning using gradient descent. Let us call $(-\infty, 6)$ the region I, and $[6, \infty)$ region II. Both solutions in Proposition 3 are constructed in such way that each one has the best fit for the points assigned to any given region, the only difference being the number of hidden units used to describe each of them. In the local optimum two neurons are used to describe region II, while only one describes region I. Symmetrically, the better solution assigns two neurons to region I (which is more complex) and only one to region II.

We believe that the core idea behind this construction can be generalized (in a non-trivial way) to high-dimensional problems. We plan to extend the construction as future work.

**Remark 3.** *In the examples we used only ReLU (and in one case a sigmoid) activation functions, as they are the most common used in practice. The similar examples can be constructed for different activation functions, however the constructions need some modifications and get more technically complicated.*

## 3.2 BAD INITIALIZATION

In this section we prove some general results regarding bad initialization phenomenon.

**Proposition 4.** *There exist an infinite amount of normalized (whitened) datasets, such that for any feed forward rectifier network architecture and an arbitrary $\epsilon \in [0, 1)$, there exists a normal distribution used to initialize the weights and biases initialized to 0 such that with probability at least $1 - \epsilon$ the gradient based techniques using log loss never achieve 0 training error nor they ever converge (gradient is never zero). Furthermore, this dataset can have a full rank covariance matrix and be linearly separable.*

*Proof.* See Appendix B.1. □

Even though the above construction requires control over the means of the normal distributions the weights are drawn from, as one can see in Figure 2, they do not have to be very large in practice. In particular, if one uses an initialization with $\sigma$ as prescribed by LeCun et al. (1998) or Glorot & Bengio (2010) then the value of $\mu = 0.24$ is sufficient to break the learning, even if we have $10,000$ hidden units in each of 100 hidden layers. Using fixed $\sigma = 0.01$ instead fails even with $\mu = 0.07$.

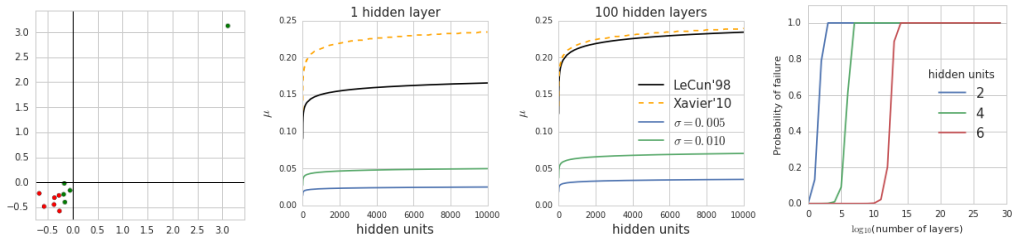

Figure 2: On the left: exemplary dataset constructed in Proposition 4, color denotes label. Two middle ones: how big the mean of the normal distribution $\mathcal{N}(\mu, \sigma^2)$ has to be in order to have at least 99% probability of the effect (very bad local minima) described in the Proposition 4, as a function of number of hidden units in $2 - h - ... - h - 1$ classification network. By LeCun'98 initialization we mean taking weights from $\mathcal{N}(\mu, \frac{1}{h})$ and by Xavier'10 from $\mathcal{N}(\mu, \frac{2}{h_{\text{in}}+h_{\text{out}}})$. In both cases the original papers used $\mu = 0$. Rightmost one: Proposition 5, probability of learning failing with increasing number of layers when the initialization is *fully correct*.

It is worth noting that even though this observations is about the existence of such dataset, our proof is actually done by *construction*, meaning that we show a way to build infinite amount of such datasets (as opposed to purely existential proofs). We would like to remark that it was well known that the initialization was important for the behaviour of learning (Glorot & Bengio, 2010; LeCun et al., 1998; Sutskever et al., 2013; Pascanu et al., 2013). Here we are exploiting these ideas in order to better understand the error surface of the model.

If we do not care about the lack of convergence, and we are simply interested in learning failure, we can prove an even stronger proposition, which works for every single dataset:

**Proposition 5.** *For every dataset, every feed forward rectifier network built for it, and every distribution used to initialize both weights and biases such that $\mathbb{E}[w] = 0, \mathbb{E}[b] = 0, \mathbf{Var}[w] > 0, \mathbf{Var}[b] \geq 0$, the probability that the gradient based training of any loss function will lead to a trivial model (predicting the same label for all datapoints) goes to 1 as the number of hidden layers goes to infinity.*

*Proof.* See Appendix B.2.  □

We can extend the previous proposition to show that for *any* regression dataset a rectifier model has at least one local minimum with a large basin of attraction (over the parameter space). Again, we rely on the *blind spots* of the rectified models. We show that there exists such *blind spot* that corresponds to a region in parameter space of same dimensionality (codimension 0). The construction relies on the fact that the dataset is finite. As such, it is bounded, and one can compute conditions for the weights of any given layer of the model such that for any datapoint all the units of that layer are deactivated. Furthermore, we show that one can obtain a better solution than the one reached from such a state. The formalization of this result is as follows.

We consider a $k$-layer deep regression model using $m$ ReLU units $\mathrm{ReLU}(x) = \max(0, x)$. Our dataset is a collection $(\mathbf{x}_i, y_i) \in \mathbb{R}^d \times \mathbb{R}$, $i = 1, \ldots, N$. We denote $\mathbf{h}_n(\mathbf{x}_i) = \mathrm{ReLU}(\mathbf{W}_n \mathbf{h}_{n-1}(\mathbf{x}_i) + \mathbf{b}_n)$ where the the ReLU functions are applied component-wise to the vector $\mathbf{W}_n \mathbf{h}_{n-1}(\mathbf{x}_i)$ and $\mathbf{h}_0(\mathbf{x}_i) = \mathbf{x}_i$. We also denote the final output of the model by $\mathcal{M}(\mathbf{x}_i) = \mathbf{W}_k \mathbf{h}_{k-1} + \mathbf{b}_k$. Solving the regression problem means finding

$$\underset{(\mathbf{W}_n)_{n=1}^k, (\mathbf{b}_n)_{n=1}^k}{\arg\min} \mathcal{L}((\mathbf{W}_n)_{n=1}^k, (\mathbf{b}_n)_{n=1}^k) = \sum_{i=1}^{N} \left[\mathcal{M}(\mathbf{x}_i) - y_i\right]^2. \quad (2)$$

Let us state two simple yet in our opinion useful Lemmata.

**Lemma 1** (Constant input). *If $\mathbf{x}_1 = \ldots = \mathbf{x}_N$, then the solution to regression (2) has a constant output $\mathcal{M} \equiv \frac{y_1 + \ldots + y_N}{N}$ (the mean of the values in data).*

*Proof.* Obvious from the definitions and the fact, that $\frac{y_1 + \ldots + y_N}{N} = \underset{c}{\arg\min} \sum_{i=1}^{N} (c - y_i)^2$.  □

**Lemma 2.** *If there holds $\mathbf{W}_1 \mathbf{x}_i < -\mathbf{b}_1$ for all $i$-s, then the model $\mathcal{M}$ has a constant output. Moreover, applying local optimization does not change the values of $\mathbf{W}_1$, $\mathbf{b}_1$.*

*Proof.* Straightforward from the definitions.  □

Combining these two lemmata yields:

**Corollary 1.** *If for any $1 \le j \le k$ there holds $\mathbf{W}_n \mathbf{h}_{n-1} < -b_n$ for all $i$-s then, after the training, the model $\mathcal{M}$ will output $\frac{y_1 + \ldots + y_N}{N}$.*

We will denote $M(\{a_1, \ldots, a_L\}) = \frac{a_1 + \ldots + a_L}{L}$ the mean of the numbers $a_1, \ldots, a_L$.

**Definition 1.** *We say that the dataset $(\mathbf{x}_i, y_i)$ is **decent** if there exists $r$ such that $M(\{y_p : \mathbf{x}_p = \mathbf{x}_r\}) \ne M(\{y_p : p = 1, \ldots, N\})$.*

**Theorem 1.** *Let $\boldsymbol{\theta} = ((\mathbf{W}_n)_{n=1}^k, (\mathbf{b}_n)_{n=1}^k)$ be any point in the parameter space satisfying $\mathbf{W}_n \mathbf{h}_n(\mathbf{x}_i) < -\mathbf{b}_n$ (coordinate-wise) for all $i$-s. Then*

   *i) $\boldsymbol{\theta}$ is a local minimum of the error surface,*

   *ii) if the first layer contains at least 3 neurons and if the dataset $(\mathbf{x}_i, y_i)$ is decent, then $\boldsymbol{\theta}$ is not a global minimum.*

*Proof.* See Appendix B.3.  □

## 4   EMPIRICAL EXAMPLES

We start our examples with experiments using MNIST dataset which show that bad initialization can lead to significant obstacles in the training process.

## 4.1 BAD INITIALIZATION ON MNIST

Figure 3 shows the training error of rectified MLP on the MNIST dataset for different seeds and different model sizes. The learning algorithm used is Adam (Kingma & Ba, 2014) and everything except initialization, when specifically stated, follows an accepted protocol (see Appendix A). The results show that models that are not initialized in a good interval do not seem to converge to a good solution of the problem even after 1,000,000 updates. Depth does not seem to be able to resolve the bad initialization of the model. The bottom row experiments are similar to those presented in Zhang et al. (2017), though more limited in their scope. They explore the correlation between the structure in the data and learning, and, at least in appearance, they do not seem to support our working hypothesis that the structure is essential. It is worth noticing though that the initialization is even more important in that setting; destroying the structure makes the model significantly more susceptible to bad initializations than when trained on the data with unpermuted labels (second column of Figure 3, the network requires at least 400 units to be able to achieve 0 training error).

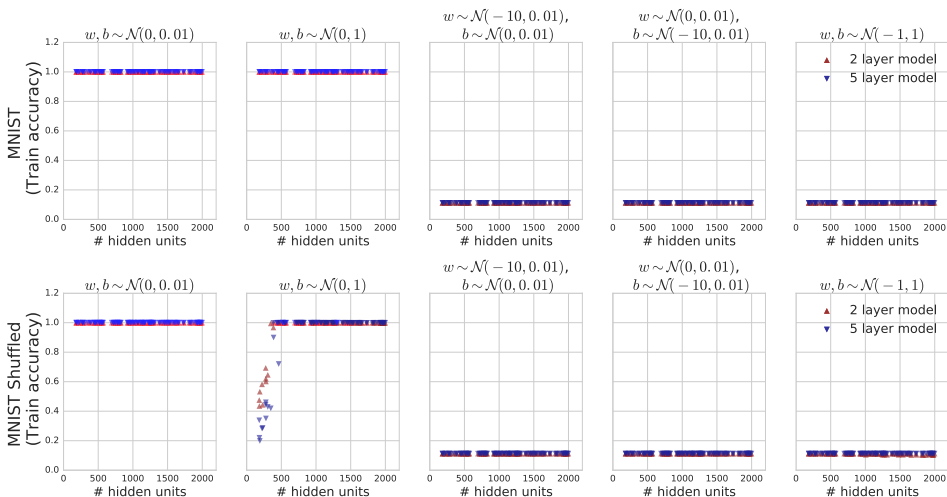

Figure 3: Plots of final training accuracy on MNIST dataset after 1,000,000 updates. Each point is a single neural net (blue triangles – 5 layer models with same number of hidden units in each layer, red triangles – 2 layer models with same number of hidden units in each layer). The title of each column shows the distribution used to initialize weights ($w$) and biases ($b$). Top row shows results on MNIST, bottom row shows results when the labels of MNIST had been randomly permuted. The number of hidden units per layer is indicated on x-axis.

The bad initializations used in these experiments are meant to target the *blind spots* of the rectifiers. The main idea is that by changing the initialization of the model (the mean of the normal distribution used to sample weights) one can force all hidden units to be deactivated for the most or for all examples in the training set. This prevents said examples from being learned, even though the task might be linearly separable. The construction may seem contrived, but it has important theoretical consequences. It shows that one can not prove well behaved learning for finite sized neural networks applied to finite sized data, without taking into account the initialization or data. We formalize this idea in the Proposition 4, making the observation that the effect can be achieved by either changing the initialization of the model, or the data. In particular, by introducing specific outliers, one can force most of the data examples in the dataset to be in the blind spot of the neural network, despite being whitened.

Details of the experimental setup are given in Appendix A. The results presented in Figure 3 (bottom row), suggest that (from an optimization perspective) the important relationship is not only the one between the inputs and targets, but also between the inputs and the way the model partitions the input space (in here we focus on rectifier models which are, from a mathematical perspective, piece-wise linear functions). To empirically test if this is a viable hypothesis we consider the MNIST dataset, where we scale the inputs by a factor $\tau$. The intuition is not to force the datasets into the *blind spot* of the model, but rather to concentrate most of the datapoints in very few linear regions (given by

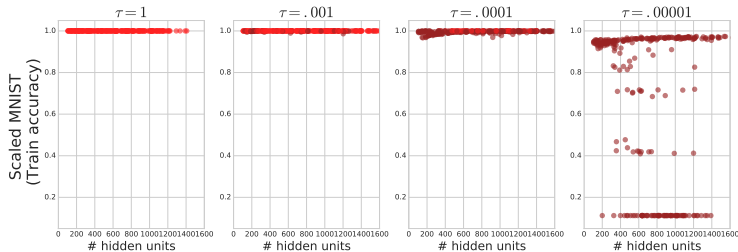

Figure 4: Plots of the final train accuracy on scaled MNIST dataset after 1,200,000 updates of a single hidden layer neural net. The title of each column shows the scaling factor applied to the data.

the initialization of the MLP). While these results do not necessarily point towards the model being locked in a bad minimum, they suggest that learning becomes less well behaved (see Fig. 4).

Additional results on a simple Zig-Zag regression task are given Figure 5. The dataset itself is in the left panel, the results are visualized in the right panel. Similarly as in the MNIST case, the experiments suggest that as data becomes more concentrated in the same linear regions (of the freshly initialized model) the learning becomes really hard, even if the model has close to 3000 units.

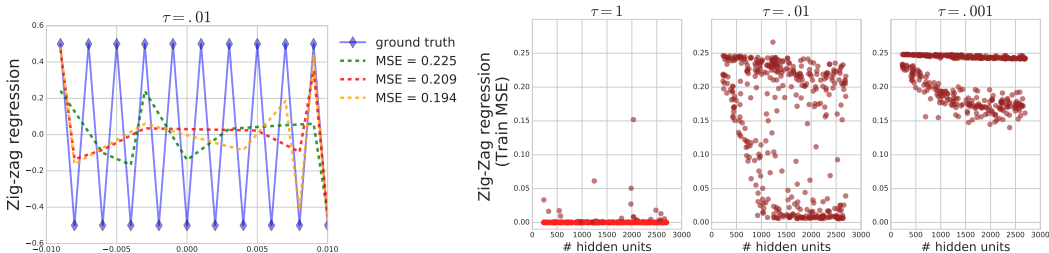

Figure 5: Plots of training MSE error on the Zig-Zag regression task after 2,000,000 updates. See caption of Figure 4 for more details. The left panel depicts the Zig-Zag regression task with three found solutions for $\tau = 0.01$. The actual datapoints are shown by the diamond shaped dots.

## 4.2 THE JELLYFISH - SUBOPTIMAL MODELS IN CLASSIFICATION USING RELU AND SIGMOIDS

To improve our understanding of learning dynamics beyond exploiting *blind spots*, we look at one of the most theoretically well-studied datasets, the XOR problem. We analyze the dataset using a single hidden layer network (with either ReLU or sigmoid units).

A first observation is that while SGD can solve the task with only 2 hidden units, full batch methods do not always succeed. Replacing gradient descent with more aggressive optimizers like Adam does not seem to help, but rather tends to make it more likely to get stuck in suboptimal solutions (Table 1).

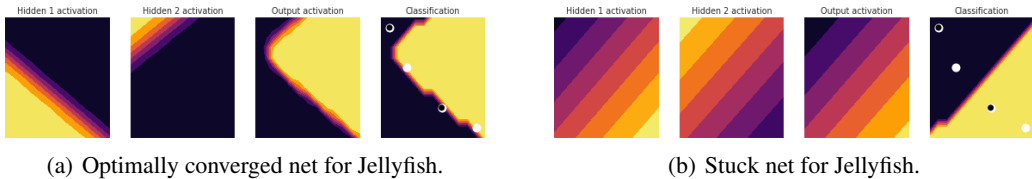

(a) Optimally converged net for Jellyfish.     (b) Stuck net for Jellyfish.

Figure 6: Examples of different outcomes of learning on the Jellyfish dataset.

By exploiting observations made in the failure modes observed for the XOR problem, we were able to construct a similar dataset, the Jellyfish, that results in suboptimal learning dynamics. The dataset

| $h$ | | XOR ReLU | XOR Sigmoid | Jellyfish ReLU | Jellyfish Sigmoid | | XOR ReLU | XOR Sigmoid | Jellyfish ReLU | Jellyfish Sigmoid |
|---|---|---|---|---|---|---|---|---|---|---|
| 2 | Adam | 28% | 79% | 7% | 0% | GD | 23% | 90% | 16% | 62% |
| 3 | Adam | 52% | 98% | 34% | 0% | GD | 47% | 100% | 33% | 100% |
| 4 | Adam | 68% | 100% | 50% | 2% | GD | 70% | 100% | 66% | 100% |
| 5 | Adam | 81% | 100% | 51% | 27% | GD | 80% | 100% | 68% | 100% |
| 6 | Adam | 91% | 100% | 61% | 17% | GD | 89% | 100% | 69% | 100% |
| 7 | Adam | 97% | 100% | 69% | 58% | GD | 89% | 100% | 86% | 100% |

Table 1: "Convergence" rate for 2-$h$-1 network with random initializations on simple 2-dimensional datasets using either Adam or Gradient Descent (GD) as an optimizer.

is formed of four datapoints, where the positive class is given by $[1.0, 0.0], [0.2, 0.6]$ and the negative one by $[0.0, 1.0], [0.6, 0.2]$. The datapoints can be seen in the Figure 6.

Compared to the XOR problem it seems the Jellyfish problem poses even more issues, especially for ReLU units, where with 4 hidden units one still only gets 2 out of 3 runs to end with 0 training error (when using GD). One particular observation (see Figure 6) is that in contrast with good solutions, when the model fails on this dataset, its behaviour close to the datapoints is almost linear. We argue hence, that the failure mode might come from having most datapoints concentrated in the same linear region of the model (in ReLU case), hence forcing the model to suboptimally fit these points.

## 5 DISCUSSION

Previous results (Dauphin et al., 2013; Saxe et al., 2014; Choromanska et al., 2015) provide insightful description of the error surface of deep models under general assumptions divorced from the specifics of the architecture. While such analysis is very valuable not only for building up the intuition but also for the development of the tools for studying neural networks, it only provides one facade of the problem. In this work we move from the generic to the specific. We show that for finite sized models/finite sized datasets one does not have a globally good behaviour of learning regardless of the model size (and even of the ratio of model size to the dataset size).

The overwhelming amount of empirical evidence points towards learning being well behaved in practice. We argue that the way to reconcile these observations is to show that the well-behaved learning dynamics are local and conditioned on the data structure, initialization and perhaps on other architectural choices. One can imagine a continuum ranging from the very specific, where every detail of the setup is important to attain good learning dynamics, to the generic, where learning is globally well behaved regardless of dataset or initialization. We believe that an important step forward in the theoretical study of the neural networks can be made by identifying where this class of models falls on that continuum. In particular, what are the most generic sets of constraints that need to be respected in order to attain the good behaviour. Our results focus on constructing counterexamples which result in a bad learning dynamics. While this does not lead directly to sufficient conditions for well-behaved systems, we hope that by carving out the space of possible conditions we are moving forward towards that goal.

Similar to Lin & Tegmark (2016) we put forward a hypothesis that the learning is only well behaved conditioned on the structure of the data. We point out, that for the purpose of learning, this structure can not be divorced from the particular initialization of the model. We postulate that learning becomes difficult if the data is structured such that there exist regions with a high density of datapoints (that belong to different classes) and the initialization results in models that assign these points to very few linear regions. While constraining the density per region alone might not be sufficient, it can provide a good starting point to understand learning for rectifier models. Another interesting question arising in that regard is what are the consequences on overfitting for enforcing a relatively low density of points per linear regions? Understanding of the structure of the error surface is an extremely challenging problem. We believe that as such, in agreement with a scientific tradition, it should be approached by gradually building up a related knowledge base, both by trying to obtain positive results (possibly under weakened assumptions, as it was done so far) and by studying the obstacles and limitations arising in concrete examples.

ACKNOWLEDGMENTS

We would want to thank Neil Rabinowitz for insightful discussions and the reviewers for their valuable remarks and suggestions.

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

## A  EXPERIMENTAL DETAILS

For the experiments depicted in Figure 3 we used MNIST dataset with data divided by 255 and one-hot encoded labels and we:

- ran 100 jobs, with the number of hidden units $h$ sampled from $[100, 2000]$ jointly for all the hidden layers, meaning that the model was $784 - h - h - ... - h - 10$,
- used Adam as an optimizer, with learning rate of $1e - 4$ (and other arguments default),
- each job was ran for 1,000,000 updates,
- batch size used was 200.

For the experiments depicted in Figure 4 we used MNIST dataset with data divided by 255 and one-hot encoded labels and we:

- ran 400 jobs, with the number of hidden units $h$ sampled from $[100, 2000]$ for MNIST experiments and $[100, 3000]$ for the Zig-Zag problem,
- we used Adam as an optimizer with learning rate randomly sampled to be either $1e - 4$ or $1e - 3$ (and other arguments default),
- each job was ran for 2,000,000 updates for the Zig-Zag problem and 1,200,000 updates in the MNIST case,
- batch size used was 10 for the Zig-Zag problem (dataset size is 20 points) and 50 for MNIST experiment.
- Initialization of weights and biases was from a Gaussian with 0 mean and standard deviation $\frac{1}{\sqrt{\text{fan}_{\text{in}}}}$.

# B  PROOFS

## B.1  PROOF OF PROPOSITION 4

For simplicity, let us consider a network with only one hidden layer of $h$ units and a binary classification task, implying we have a single output neuron. Analogous result holds for any number of classes and for any arbitrary depth of the network as well. We use the following notation: $\mathbf{h}(\mathbf{x}_i)$ is a vector of activations of hidden units when presented with $i$-th training sample, $\mathcal{M}(\mathbf{x}_i)$ is the activation of the output neuron given the same sample, $\mathbf{W}$ is matrix of hidden weights, $\mathbf{b}$ is a vector of biases in the hidden layer and finally $\mathbf{v}, c$ are the weights and bias in the output layer. The whole classification becomes

$$\mathcal{M}(\mathbf{x}_i) = \mathbf{v}\mathrm{ReLU}(\mathbf{W}\mathbf{x}_i + \mathbf{b}) + c.$$

Let us now consider a dataset where $N-1$ points have all features negative, and a single point, denoted $\mathbf{x}_{i^*}$ (with positive label) which has all the features positive. We can always find such points that their coordinate-wise mean is equal to $0$ and their standard deviation equals $1$, since we can place all $N-1$ points very close to the origin, and the point $\mathbf{x}_{i^*}$ arbitrary far away in the positive part of the input space. Our dataset is therefore normalized (whitened) and it can have a full rank covariance matrix (since the construction does not depend on nothing besides signs of the features). We want to compute

$$P(\forall_{i \neq i^*}\mathbf{h}(\mathbf{x}_i) = 0 \wedge \mathbf{h}(\mathbf{x}_{i^*}) > 0).$$

Since, by construction $\forall_{i \neq i^*}\mathbf{x}_i < 0$ and $\mathbf{x}_{i^*} > 0$, it is also true that if all the weights $\mathbf{W}$ are non-negative (and at least one is positive), then all the activations (after ReLU) of the hidden units will be $0$ besides the one positive activation of $\mathbf{h}(\mathbf{x}_{i^*})$ which comes directly from the assumption that biases are initialized to $0$[1]. Consequently

$$P\left(\forall_{i \neq i^*}\mathbf{h}(\mathbf{x}_i) = 0 \wedge \mathbf{h}(\mathbf{x}_{i^*}) > 0\right) \geq P(\mathbf{W} > 0),$$

and given that weights initializations are independent samples from the same distribution we get

$$P(\mathbf{W} > 0) = \prod_{i=1}^{dh} \int_0^\infty \mathcal{N}(\mu, \sigma^2) = \left(\int_0^\infty \mathcal{N}(\mu, \sigma^2)\right)^{dh},$$

where $\mu, \sigma$ are parameters of the distribution we use to initialize weights and $d$ is input space dimensionality. All that is left is to show that during any gradient based optimization these weights will not be corrected, which requires one more assumption - that the output weights are positive as well. If this is true, then $\forall_{i \neq i^*}\mathcal{M}(\mathbf{x}_i) = 0$ (again using that the output bias is zero) and $\mathcal{M}(\mathbf{x}_{i^*}) > 0$ and of course $P(\mathbf{v} > 0) = (\int_0^\infty \mathcal{N}(\mu, \sigma^2))^h$. Now we have a fully initialized model which maps all the samples to $0$, and one positive sample to some positive value. Consequently, given the fact that we use log loss, there holds $\partial \mathcal{L}/\partial v_k > 0$ and $\partial \mathcal{L}/\partial w_{kl} \geq 0$ for all $k, l$. Indeed, since these changes are all increasing the probability of good classification of $\mathbf{x}_{i^*}$ and all remaining points are in inactive part of ReLUs thus they cannot contribute to partial derivatives. Therefore, during any gradient based (as well as stochastic and mini-batch based) learning the projection of samples mapped to $0$ will not change, and the projection of $\mathbf{x}_{i^*}$ will grow to infinity (so the sigmoid approaches $1$). Consequently we constructed an initialization scheme which with probability at least

$$P(\mathbf{W}, \mathbf{v} > 0) = P(\mathbf{W} > 0)P(\mathbf{v} > 0) = \left(\int_0^\infty \mathcal{N}(\mu, \sigma^2)\right)^{dh+h} = 1 - \epsilon, \qquad (3)$$

gives the initial conditions of the net, where despite learning with log loss, we always classify all, arbitrary labeled $N-1$ points to the same label (since they are all mapped to the same output value) and we classify the unique positive sample with the valid label. Furthermore - the optimization never finishes, and there exists a network with a better accuracy, as we can label $N-1$ points in any manner, including making it *linearly separable* by labeling according to the first feature only.

In order to generalize to arbitrary number of layers, we would similarly force all the weights to be positive, thus with parametrization of $k$-layer deep network $\boldsymbol{\theta} = (\mathbf{W}_n)_{n=1}^k$ we would simply get

$$P(\forall_{1 \leq n \leq k}\mathbf{W}_n > 0) = \prod_{n=1}^k P(\mathbf{W}_n > 0) = \left(\int_0^\infty \mathcal{N}(\mu, \sigma^2)\right)^{dh+(k-2)h^2+h} = 1 - \epsilon,$$

---

[1]We can still construct similar proof for biases taken from normal distributions as well.

and finally, if the biases are not zero, but have some some arbitrary values (fixed or sampled) we simply adjust the size of the weights accordingly. Instead of having them bigger than $0$ we would compute probability of having them big enough to make the whole first layer to produce 0s for every point besides $\mathbf{x}_{i*}$ and analogously for the remaining layers.

Furthermore, it is possible to significantly increase the probability of failure if we do not mind the situation in which the learning process is not even starting. Proposition 5 addresses this situation.

## B.2 PROOF OF PROPOSITION 5

*Proof.* Let us notice, that since we are using ReLU activations, the activations on $j$th hidden layer $\mathbf{h}_j$ are non-negative. Consequently, if the following layer has only negative weights and non-positive biases, then $\mathbf{h}_{j+1} \equiv \mathbf{b}_{j+1}$ (as all the ReLUs are inactive), so the network has to output exactly the same value for every single point. Furthermore, during the gradient based learning we will never change those weights as gradients going through ReLUs will be equal to zero.

Let us now consider a deep rectifier network with $k$ hidden layers with $h$ neurons each. If only $k > 2$ we can use the above observation to compute for every $j > 1$:

$$P(\text{All neurons inactive in hidden layer } j) \geq P(\mathbf{b}_j \leq 0, \mathbf{W}_j < 0) = P(\mathbf{b}_j \leq 0)P(\mathbf{W}_j < 0),$$

and consequently, due to the assumptions about expected values

$$
\begin{aligned}
P(\text{Learning fails with } k \text{ hidden layers}) &\geq P(\exists_{j>1}\text{All neurons inactive in hidden layer } j) \\
&\geq P(\exists_{j>1}\forall_{l,i} b_{jl} \leq 0 \wedge W_{ji} < 0) \\
&= 1 - P(\forall_{j>1}\exists_l b_{jl} > 0 \vee \exists_i W_{ji} \geq 0) \\
&= 1 - \prod_{j=1}^{k-1} P(\exists_l b_{jl} > 0 \vee \exists_i W_{ji} \geq 0)
\end{aligned}
$$

As every layer has the same number $h$ of neurons, the values $(\exists_l b_{jl} > 0 \vee \exists_i W_{ji} \geq 0)$ are equal for every $j$. Therefore

$$
\begin{aligned}
1 - \prod_{j=1}^{k-1} P(\exists_l b_{jl} > 0 \vee \exists_i W_{ji} \geq 0) &= 1 - P(\exists_l b_{1l} > 0 \vee \exists_i W_{1i} \geq 0)^{k-1} \\
&= 1 - (1 - P(\forall_i b_{1i} \leq 0 \wedge W_{1i} < 0))^{k-1} \\
&= 1 - [1 - \text{cdf}_{\mathbf{b}}(0)^h \text{cdf}_{\mathbf{W}}(0)^{h^2}]^{k-1}.
\end{aligned}
$$

Due to assumptions about distributions of biases and weights we know that

$$0 < \text{cdf}_{\mathbf{W}}(0) < 1, 0 \leq \text{cdf}_{\mathbf{b}}(0) < 1$$

so

$$\lim_{k \to \infty} 1 - [1 - \text{cdf}_{\mathbf{b}}(0)^h \text{cdf}_{\mathbf{W}}(0)^{h^2}]^{k-1} = 1.$$

$\square$

## B.3 PROOF OF THEOREM 1

Claim i) is a direct consequence of Corollary 1. It remains to prove ii). For that it is sufficient to show an example of a set of weighs $\hat{\theta} = ((\hat{\mathbf{W}}_n)_{n=1}^k, (\hat{\mathbf{b}}_n)_{n=1}^k)$ such that $\mathcal{L}((\mathbf{W}_n)_{n=1}^k, (\mathbf{b}_n)_{n=1}^k) > \mathcal{L}((\hat{\mathbf{W}}_n)_{n=1}^k, (\hat{\mathbf{b}}_n)_{n=1}^k)$. Let $r$ be such that $M(\{y_p : \mathbf{x}_p = \mathbf{x}_r\}) \neq M(\{y_p : p = 1, \ldots, N\})$. Such point exists by assumption that the dataset is decent. Let $\mathcal{H}$ be a hyperplane passing through $\mathbf{x}_r$ such that none of the points $\mathbf{x}_s \neq \mathbf{x}_r$ lies on $\mathcal{H}$. Then there exists a vector $\mathbf{v}$ such that $|\mathbf{v}^T(\mathbf{x}_s - \mathbf{x}_r)| > 2$ for all $\mathbf{x}_s \neq \mathbf{x}_r$. Let $\gamma = \mathbf{v}^T \mathbf{x}_r$. We define $\mathbf{W}_1$ in such a way that the first row of $\mathbf{W}_1$ is $\mathbf{v}$ , the second row is $2\mathbf{v}$ and the third one is $\mathbf{v}$ again, and if the first layer has more than 3 neurons, we put all the remaining rows of $\mathbf{W}_1$ to be equal zero. We choose the first three biases of $\mathbf{b}_1$ to be $-\gamma + 1$, $-2\gamma$ and $-\gamma - 1$ respectively. We denote $\mu = M(\{y_p : \mathbf{x}_p \neq \mathbf{x}_r\})$ and $\nu = M(\{y_p : \mathbf{x}_p = \mathbf{x}_r\})$. We then choose $\mathbf{W}_2$ to be a matrix whose first row is $(\nu - \mu, \mu - \nu, \nu - \mu, 0, \ldots, 0)$ and the other

rows are equal to 0. Finally, we choose the bias vector $\mathbf{b}_2 = (\mu, 0, \ldots, 0)^T$.
If our network has only one layer the output is

$$(\nu - \mu)\mathrm{ReLU}(\mathbf{v}^T \mathbf{x}_p - \gamma + 1) - (\nu - \mu)\mathrm{ReLU}(2\mathbf{v}^T \mathbf{x}_p - 2\gamma) + (\nu - \mu)\mathrm{ReLU}(\mathbf{v}^T \mathbf{x}_p - \gamma - 1) + \mu.$$

For every $\mathbf{x}_p = \mathbf{x}_r$ this yields $(\nu - \mu) \cdot 1 - 0 + 0 + \mu = \nu$. For any $\mathbf{x}_p \neq \mathbf{x}_r$ we either have $\mathbf{v}^T \mathbf{x}_p - \gamma < -2$ yielding $0 - 0 + 0 + \mu = \mu$ or $\mathbf{v}^T \mathbf{x}_p - \gamma > 2$ yielding $(\nu - \mu)(\mathbf{v}^T \mathbf{x}_p - \gamma + 1) - (\nu - \mu)(2\mathbf{v}^T \mathbf{x}_p - 2\gamma) + (\nu - \mu)(\mathbf{v}^T \mathbf{x}_p - \gamma - 1) + \mu = \mu$.
In case the network has more than 1 hidden layer we set all $\mathbf{W}_n = \mathbf{I}$ (identity matrix) and $\mathbf{b}_n = \mathbf{0}$ for $n = 3, \ldots, k$.
If we denote $\bar{\mu} = M(\{y_p : p = 1, \ldots, N\})$ (mean of all labels), we get:

$$\mathcal{L}((\hat{\mathbf{W}}_n)_{n=1}^k, (\hat{\mathbf{b}}_n)_{n=1}^k) = \sum_{\mathbf{x}_p \neq \mathbf{x}_r} (y_i - \mu)^2 + \sum_{\mathbf{x}_p = \mathbf{x}_r} (y_i - \nu)^2 <$$

$$\sum_{\mathbf{x}_p \neq \mathbf{x}_r} (y_i - \bar{\mu})^2 + \sum_{\mathbf{x}_p = \mathbf{x}_r} (y_i - \bar{\mu})^2 = \sum_{y_i}(y_i - \bar{\mu})^2 = \mathcal{L}((\mathbf{W}_n)_{n=1}^k, (\mathbf{b}_n)_{n=1}^k).$$

We used the fact that for any finite set $A$ the value $M(A)$ is a strict minimum of $f(c) = \sum_{a \in A}(a - c)^2$ and the assumption that $\nu \neq \bar{\mu}$.

## B.4  PROOF OF PROPOSITION 1

There holds $\mathcal{L}(1, -3, 1, 0) = 0+0+0+9+9 = 18$, and $\mathcal{L}(-7, -4, 1, 0) = 4+1+0+9+0 = 14$, thus $(1, -3, 1, 0)$ cannot be a global minimum. It remains to prove that $(1, -3, 1, 0)$ is a local minimum, i.e. that $\mathcal{L}(1 + \delta_w, -3 + \delta_b, 1 + \delta_v, \delta_c) \geq \mathcal{L}(1, -3, 1, 0)$ for $|\delta_w|, |\delta_b|, |\delta_v|, |\delta_c|$ sufficiently small. We need to consider two cases:
**ReLU activated at** $x_3$. In that case

$$\mathcal{L}(1 + \delta_w, -3 + \delta_b, 1 + \delta_v, \delta_c) =$$

$$((1 + \delta_v)(3 + 3\delta_w - 3 + \delta_b) + \delta_c)^2 + ((1 + \delta_v)(4 + 4\delta_w - 3 + \delta_b) + \delta_c - 1)^2 +$$

$$((1 + \delta_v)(5 + 5\delta_w - 3 + \delta_b) + \delta_c - 2)^2 + (\delta_c + 3)^2 + (\delta_c - 3)^2.$$

We introduce new variables $x = (\delta_w + 1)(1 + \delta_v) - 1$, $y = (\delta_b - 3)(1 + \delta_v) + 3$, $z = \delta_c$. The formula becomes

$$(3x + y + z)^2 + (4x + y + z)^2 + (5x + y + z)^2 + 2z^2 + 18 \geq 18,$$

which ends the proof in this case.
**ReLU deactivated at** $x_3$. In that case

$$\mathcal{L}(1 + \delta_w, -3 + \delta_b, 1 + \delta_v, \delta_c) = \delta_c^2 + ((1 + \delta_v)(4 + 4\delta_w - 3 + \delta_b) + \delta_c - 1)^2 +$$

$$((1 + \delta_v)(5 + 5\delta_w - 3 + \delta_b) + \delta_c - 2)^2 + (\delta_c + 3)^2 + (\delta_c - 3)^2 =$$

$$(4x + y + z)^2 + (5x + y + z)^2 + 3z^2 + 18 \geq 18$$

(we used $x = (\delta_w + 1)(1 + \delta_v) - 1$, $y = (\delta_b - 3)(1 + \delta_v) + 3$, $z = \delta_c$ again).
Note that due to the assumption that $|\delta_w|, |\delta_b|, |\delta_v|, |\delta_c|$ are sufficiently small the ReLU is always activated at $x_1, x_2$ and deactivated at $x_4, x_5$.

