# Peer review of "Local minima in training of deep networks"

_ICLR 2017 — rejected_

[Official Review · AnonReviewer1 · rating 5 · confidence 4 · 20 Dec 2016]
**Presents interesting cases of local minima in neural networks, but contains technical issues**

This paper studies the error surface of deep rectifier networks, giving specific examples for which the error surface has local minima. Several experimental results show that learning can be trapped at apparent local minima by a variety of factors ranging from the nature of the dataset to the nature of the initializations. This paper develops a lot of good intuitions and useful examples of ways that training can go awry. 

Even though the examples constructed in this paper are contrived, this does not necessarily remove their theoretical importance. It is very useful to have simple examples where things go wrong. However the broader theoretical framing of the paper appears to be going after a strawman.

“The underlying easiness of optimizing deep networks does not simply rest just in the emerging structures due to high dimensional spaces, but is rather tightly connected to the intrinsic characteristics of the data these models are run on.” I believe this perspective is already contained in several of the works cited as not belonging to this perspective. Choromanska et al., for instance, analyze Gaussian inputs, and so clearly make claims based on characteristics of the data the models are run on. More broadly, the loss function is determined jointly by the dataset and the model parameters, and so no account of the error surface can be separated from dataset properties. It is not clear to me what ‘emerging structures due to high dimensional spaces’ are, or what they could be, that would make them independent of the dataset and initial model parameters. The emerging structure of the error surface is necessarily related to the dataset and model parameters.

Again, a key worry with this paper is that it is aiming at a strawman: replica methods characterize average behavior for infinite systems, so it is not surprising that specific finite sized systems might yield poor optimization landscapes. The paper seems surprised that training can be broken with a bad initialization, but initialization is known to be critical, even for linear networks: saddle points are not innocuous, with bad initializations dramatically slowing learning (e.g. Saxe et al. 2014).

It seems like the proof of proposition 5 may have an error. Suppose cdf_b(0) = 0 and cdf_W(0)=1/2. We have P(learning fails) >= 1 - 1/2^{h^2(k-1)}, meaning that the probability of failure _increases_ as the number of hidden units increases. It seems like it should rather be (ignoring the bias) p(fails) >= 1 - [ 1 - p(w<0)^h^2]^{k-1}. In this case the limit as k-> infinity depends on how h scales with k, so it is no longer necessarily true that “one does not have a globally good behaviour of learning regardless of the model size.”

The paper also appears to insufficiently distinguish between local minima and saddle points. Section 3.1 states it shows training being stuck in a local minimum, but this is based on training with a fixed budget of epochs. It is not possible to tell whether this result reflects a genuine local minimum or a saddle point based on simulation results. 
It may also be the case that, while rectifiers suffer from genuine blind spots, sigmoid or soft rectifier nonlinearities may not. On the XOR problem with two hidden nodes, for instance, it was thought that were local minima but in fact there are none (e.g. L. Hamey, “Analysis of the error surface of the XOR network with two hidden nodes,” 1995). 

If the desire is simply to show that training does not converge for particular finite problems, much simpler counterexamples can be constructed and would suffice: set all hidden unit weights to zero, for instance. 

In the response to prereview questions, the authors write ‘If the “complete characterization” [of the error surface] was indeed universally valid, we would not be able to break the learning with the initialization’ but, as mentioned previously, the basic results for even deep linear networks show that a bad initialization (at or near a saddle point) will break learning. Again, it seems this paper is attacking a straw man along the lines of “nothing can possibly go wrong with neural network training.” No prior theoretical result claims this. 

The Figure 2 explanation seems counterintuitive to me. Simply scaling the input, if the weight matrices are initialized with zero biases, will not change the regions over which each ReLU activates. That is, this manipulation does not achieve the goal of concentrating “most of the data points in very few linear regions.” A far more likely explanation is that the much weaker scaling has not been compensated by the learning algorithm, but the algorithm would converge if run longer. The response notes that training has been conducted for an order of magnitude longer than required for the unscaled input to converge, but the scaling on the data is not one but five orders of magnitude—and indeed the training does converge without issue for scaling up to four orders of magnitude. The response notes that Adam should compensate for the scaling factor, but this depends on the details of the Adam implementation—the epsilon factor used to protect against division by zero, for example. 

This paper contains many interesting results, but a variety of small technical concerns remain.

[Official Review · AnonReviewer2 · rating 3 · confidence 5 · 20 Dec 2016]
**analyzed models and datasets are not typical of those encountered in practice**

The paper studies some special cases of neural networks and datasets where optimization fails. Most of the considered models and datasets are however highly constructed and do not follow the basic hyperparameters selection and parameter initialization heuristics. This reduces the practical relevance of the analysis.

The experiment "bad initialization on MNIST" shows that for very negative biases or weights drawn from a non-centered distribution, all ReLU activations are "off" for all data points, and thus, optimization is prevented. This never occurs in practice, because using proper initialization heuristics avoid these cases.

The "jellyfish" dataset constructed by the authors is demonstrated to be difficult to fit by a small model. However, the size/depth of the considered model is unsuitable for this problem.

Proposition 4 assumes that we can choose the mean from which the weight parameters are initialized. This is typically not the case in practice as most initialization heuristics draw weight parameters from a distribution with mean 0.

Proposition 5 considers infinitely deep ReLU networks. Very deep networks would however preferably be of type ResNet.

[Official Review · AnonReviewer3 · rating 5 · confidence 3 · 21 Dec 2016]

The main merit of this paper is to draw again attention to how crucial initialization of deep network *can* be; and to counter the popular impression that modern architectures and improved gradient descent techniques make optimization local minima and saddle points no longer a  problem. 

While the paper provides interesting counter-examples that showcase how bad initialization mixed with particular data can lead the optimization to get stuck at a poor solution, these feel like contrived artificial constructs. More importantly the paper does not consider popular heuristics that likely help to avoid getting stuck, such as: non-saturating activation functions (e.g. leaky RELU), batch-norm, skip connections (resnet), that can all be thought of as contributing to keep the gradients flowing. 

The paper puts up a big warning sign about potential initialization problems (with standard RELU nets), but without proposing new solutions or workarounds, nor carrying out a systematic analysis of how this picture is affected by most commonly used current heuristic techniques (in architecture, initialization and training). Such a broader scope analysis, especially if it did lead to insights of practical relevance, could much increase the value of the paper for the reader.

[Final Decision · Program Chairs · 06 Feb 2017]
**ICLR committee final decision**

This paper analyzes under which circumstances bad local optima prevent effective training of deep neural networks. The contribution is real, but the gap between the proposed scenarios and real training scenarios diminishes its importance.